# Exploring the influence of competition on arbovirus invasion risk in communities

**Afonso Dimas Martins**[ID]**[1]\***, **Quirine ten Bosch**[ID]**[2]**, **J. A. P. Heesterbeek**[ID]**[1]**

**1** Department of Population Health Sciences, Faculty of Veterinary Medicine, University of Utrecht, Utrecht, The Netherlands, **2** Quantitative Veterinary Epidemiology, Wageningen University and Research, Wageningen, The Netherlands

\* a.l.moreiradimasmartins@uu.nl

## Abstract

Arbovirus outbreaks in communities are affected by how vectors, hosts and non-competent species interact. In this study, we investigate how ecological interactions between species and epidemiological processes influence the invasion potential of a vector-borne disease. We use an eco-epidemiological model to explore the basic reproduction number $R_0$ for a range of interaction strengths in key processes, using West Nile virus infection to parameterize the model. We focus our analysis on intra and interspecific competition between vectors and between hosts, as well as competition with non-competent species. We show that such ecological competition has non-linear effects on $R_0$ and can greatly impact invasion risk. The presence of multiple competing vector species results in lower values for $R_0$ while host competition leads to the highest values of risk of disease invasion. These effects can be understood in terms of how the competitive pressures influence the vector-to-host ratio, which has a positive relationship with $R_0$. We also show numerical examples of how vector feeding preferences become more relevant in high competition conditions between hosts. Under certain conditions, non-competent hosts, which can lead to a dilution effect for the pathogen, can have an amplification effect if they compete strongly with the competent hosts, hence facilitating pathogen invasion in the community.

## Introduction

Vector-borne diseases are infections that result from pathogens that are transmitted by vectors to hosts. Mosquito vectors feed on animals as they require blood for completing their life cycle. During feeding, pathogens may be transmitted, causing infections in suitable hosts. West Nile Virus (WNV) is a neuroinvasive vector-borne disease transmitted by mosquitoe species, mainly of the *Culex* genus, to a range of animals, such as birds, horses, and humans. The symptoms in humans range from headaches and fever to death in critical situations [1]. The virus is maintained in circulation through enzootic amplification in transmission cycles between mosquito species and bird species. Other species, such as horses and humans, are only incidental hosts and are non-competent for the virus, i.e., they do not produce enough viremia to further contribute to transmission [1].

**Funding:** This work is included in the research programme One Health PACT (project number 109986), which is partly financed by the Dutch Research Council (NWO). The funders had no role in study design, data collection and analysis, decision to publish, or preparation of the manuscript.

**Competing interests:** The authors have declared that no competing interests exist.

The dynamics of a pathogen in multi-species communities depend on the transmission routes available, as well as on the direct and indirect ecological interactions within and between species [2–4]. One type of interaction is competition, which in the case of vector-borne diseases can occur at different levels: between the vectors, the hosts, and potentially also between different pathogens. These competitive forces are characterized in different ways depending on the species involved. For example, at the level of the vectors, mosquitoes compete mostly during early larval stages when the resources are more limited [5]. At the host level, birds compete mostly for food, territory, and mating opportunities [6, 7]. At the within-host level, pathogens can interact, for example, by releasing toxic compounds or otherwise interfere with the growth of their competitors [8]. All the species in their different roles affect the others in some way: pathogens are dependent on the vectors and the vertebrate hosts to be maintained in transmission; vectors rely on hosts for blood-feeding; and hosts suffer from fitness loss due to the vector-transmitted infections. Therefore, any changes in the species abundances or interaction strengths in the community can influence the network of pathogen transmission, and have impact on the probability of pathogen emergence.

Host species differ in their competence [9] with respect to the pathogen. Species that we will collectively refer to as 'non-competent species' abound in the natural habitat of any pathogen, for example the dead-end host species and species that are not susceptible. Such non-competent species interact ecologically with host species of various degrees of competence and influence the abundance and dynamics of host species. Non-competent species may therefore influence pathogen dynamics indirectly and also affect invasion risk upon introduction, potentially both positively and negatively. For vector-borne pathogens, additional interactions become relevant, for example related to blood meals necessary for completion of the vector life cycle. This may involve host species of different competence with respect to the pathogen in question, as well as dead-end host species and species not susceptible to the pathogen. It is known that vector-borne pathogens are sensitive to biodiversity [10], however whether pathogens end up benefiting from changes in biodiversity depends on many factors [11]. For example, non-competent species or very weakly competent host species can dilute mosquito bites on competent host species. These bites do not contribute to maintaining the pathogen cycle. It is, therefore, important to explore a more diverse collection of species in a community to better understand outbreak risk in complex natural environments.

Our aim here is to show that ecological interactions may influence host, vector and pathogen dynamics in commonly adopted modelling descriptions of mosquito-borne pathogen systems. We focus on invasion dynamics. In this study, we use WNV as a motivating case study for the exploration of invasion risk of a mosquito-borne virus in a multi-species setting. Like many other vector-borne infections, WNV is expanding its range, with small outbreaks in areas that are at the borders of the current endemic range. For such small outbreaks it is important to understand how local conditions can be conducive to onward transmission. For these reasons, we focus in this study on invasion risk, rather than on long-term dynamics. We construct a low-dimensional model that captures explicit competition acting at different levels. The focus is on intra- and interspecific competition as the main ecological interaction. Marini et al. 2017 [12], for example, studied the effects of competition at the host level on the invasion risk of WNV. In our study, we include not only competition between hosts, but also competition between mosquito species. This is also further extended to more complex scenarios such as the presence of non-competent vectors and non-competent vertebrate species, and their respective competitive pressures on the competent species. This approach allows us to study the potential effects of species interactions that are difficult to quantify in the field, such as competition at different trophic levels and vector feeding preferences. The invasion risk is quantified by the basic reproduction number $R_0$ [13]. Because our interest is in showing the

effects of ecological interactions in common settings for the pathogen, we consider situations where, apart from those interactions, there are no impediments to invasion. The steps taken here to obtain insight into $R_0$ can be generalized to other systems with multiple vector, host and non-host species, as well as to additional types of ecological interaction.

## Materials and methods

### Infection in a multi-vector and multi-host community

We constructed a compartmental model for a simplified vector-borne disease with an enzootic cycle. We followed the methods of [13] and extended the approach of [12], based on assumptions fitting to our aims. We consider the invasion of WNV in a community with two mosquito subspecies of the *Culex* genus, *Culex pipiens*, $N_1$, and *Culex molestus*, $N_2$, and two bird species, the American crow (*Corvus brachyrhynchos*), $N_3$, and the House finch (*Haemorhous mexicanus*), $N_4$. Our choice of species was based on the amount of data available for these species, thus minimizing the gaps in parameterization for a common setting where WNV can invade. Key life-history mechanisms for these species, such as birth rates and transmission probabilities, were used to inform parameter values (Table 1). In some cases simplifications were made, for example, biting rates are not constant and depend on environmental factors such as temperature. Since we are considering only the relatively short timescale of disease invasion, we use point estimates rather than functions for each parameter. We assume a closed population with abundances at the infection-free steady state derived before (i.e., $N_i = N_i^*$, for $i \in \{1, 2, 3, 4\}$). There are four equations governing the dynamics of the population size of each

**Table 1. Parameter definitions and values used in the numerical simulations.** All rates shown are per day. The competition coefficients $c_{ii}$, $c_{ij}$ and feeding preferences $\lambda_i$ are varied in the numerical simulations and otherwise fixed to these default values.

| Parameter | Description | Estimate | References |
|---|---|---|---|
| $v_1$, $v_2$ | Vector birth rates | 0.537 | [18] |
| $v_3$, $v_4$ | Host birth rates | 0.5 | [19] |
| $\mu_1$, $\mu_2$ | Vector death rate | 0.08 | [17] |
| $\mu_3$, $\mu_4$ | Host death rate | $2.7 \times 10^{-4}$ | [20] |
| $\alpha_3$ | Additional disease-related death rate in American crow | 0.2 | [16] |
| $\alpha_4$ | Additional disease-related death rate in House finch | 0.11 | [16] |
| $c_{ii}$ | Intraspecific competition in species $i$ | 0.05 | Assumption |
| $c_{ij}$ | Interspecific competition of species $j$ on species $i$ | 0.1 | Assumption |
| $d_{ii}$ | Fraction of competition from $i$ that affects its own death rate | 0.1 | Assumption |
| $d_{ij}$ | Fraction of competition from $j$ that affects the death rate of $i$ | 0.2 | Assumption |
| $q_1$, $q_2$ | Vertical transmission probability | 0.004 | [17, 21] |
| $p_{13}$, $p_{14}$ | Transmission probability bird to *C. pipiens* | 0.09[a] | [22] |
| $p_{23}$, $p_{24}$ | Transmission probability bird to *C. molestus* | 0.13[a] | [22] |
| $p_{31}$, $p_{41}$ | Transmission probability *C. pipiens* to bird | 0.80 | [17, 23] |
| $p_{32}$, $p_{42}$ | Transmission probability *C. molestus* to bird | 0.80 | [17, 23] |
| $p_{33}$ | Horizontal transmission probability | 0.83 | [16] |
| $b_1$ | Biting rate of *C. pipiens* on bird | 0.4 | [24] |
| $b_2$ | Biting rate of *C. molestus* on bird | 0.4 | [24] |
| $b_3$ | Contact rate between crows | 0.2 | [16] |
| $\lambda_i$ | Feeding preference of vector $N_i$ towards a given host | 1 | Assumption |

[a] In this study the authors estimated this parameter for three different temperatures (18º C, 23º C and 28º C). The point estimate for 23º C was the assumed value for this parameter.

species, $N_i = S_i + I_i$, i.e. the sum of the susceptible and infectious sub-populations of species $i$, and an additional four equations responsible for the dynamics of the infectious sub-populations, $I_i$. The populations of both the vector and host species are assumed to follow Lotka-Volterra dynamics, with intraspecific and interspecific competition at both the vector species and the host species level (Fig 1a).

The system is given by

$$
\begin{aligned}
N_1' &= v_1 N_1 - (c_{11} N_1 + c_{12} N_2 + \mu_1) N_1 \\
N_2' &= v_2 N_2 - (c_{21} N_1 + c_{22} N_2 + \mu_2) N_2 \\
N_3' &= v_3 N_3 - (c_{33} N_3 + c_{34} N_4 + \mu_3) N_3 - \alpha_3 I_3 \\
N_4' &= v_4 N_4 - (c_{43} N_3 + c_{44} N_4 + \mu_4) N_4 - \alpha_4 I_4 \\
I_1' &= q_1 (v_1 - (1 - d_{11}) c_{11} N_1 - (1 - d_{12}) c_{12} N_2) I_1 \\
&\quad + \left( \beta_{13} \frac{\lambda_1 I_3}{\lambda_1 N_3 + N_4} + \beta_{14} \frac{I_4}{\lambda_1 N_3 + N_4} \right) S_1 - (c_{11} d_{11} N_1 + c_{12} d_{12} N_2 + \mu_1) I_1 \\
I_2' &= q_2 (v_2 - (1 - d_{21}) c_{21} N_1 - (1 - d_{22}) c_{22} N_2) I_2 \\
&\quad + \left( \beta_{23} \frac{\lambda_2 I_3}{\lambda_2 N_3 + N_4} + \beta_{24} \frac{I_4}{\lambda_2 N_3 + N_4} \right) S_2 - (c_{21} d_{21} N_1 + c_{22} d_{22} N_2 + \mu_2) I_2 \\
I_3' &= \left( \beta_{31} \frac{\lambda_1 I_1}{\lambda_1 N_3 + N_4} + \beta_{32} \frac{\lambda_2 I_2}{\lambda_2 N_3 + N_4} + \beta_{33} \frac{I_3}{N_3} \right) S_3 \\
&\quad - (c_{33} d_{33} N_3 + c_{34} d_{34} N_4 + \mu_3 + \alpha_3) I_3 \\
I_4' &= \left( \beta_{41} \frac{I_1}{\lambda_1 N_3 + N_4} + \beta_{42} \frac{I_2}{\lambda_2 N_3 + N_4} + \beta_{44} \frac{I_4}{N_4} \right) S_4 \\
&\quad - (c_{43} d_{43} N_3 + c_{44} d_{44} N_4 + \mu_4 + \alpha_4) I_4
\end{aligned}
\tag{1}
$$

with all parameters defined in Table 1. The system is based on the following assumptions:

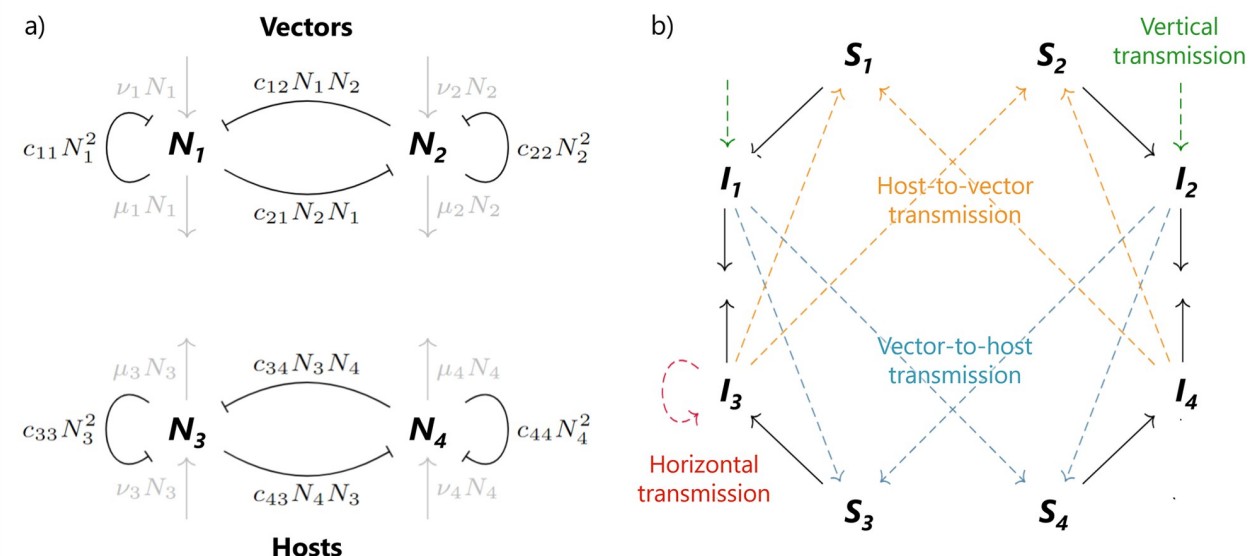

**Fig 1. Diagrams of the ecological (a) and epidemiological (b) components of the model.** Gray arrows: demography related movements, t-bars: competition, black arrows: infection related transitions, dashed arrows: transmission routes (green: vertical, yellow: host-to-vector, blue: vector-to-host, red: horizontal).

- Constant birth rate $v_i$ and constant (natural) death rate $\mu_i$ for each species. The births of the vectors are, therefore, independent of the abundances of host species $N_3$ and $N_4$. Effectively, we assume that the abundances of host species 3 and 4 are such that they do not limit vector feeding. That is, they are present in such numbers that all vectors can become satiated. We consider infection-induced host mortality by adding the parameter $\alpha_i$, an additional death rate caused by the infection, and assume that the vectors do not suffer from pathogen-related mortality.

- We include competition between the vector species (both within and between-species competition), as well as competition between the host species (both within and between-species competition), see Fig 1a). All types of competition are described by the parameters $c_{ij}$ and $d_{ij}$, as outlined in more detail below.

- We assume an SI-type epidemiological dynamics, meaning that infectivity is life-long and there is no latency period. For mosquitoes the latter is a strong assumption given their relatively short lifespan. We show in the Supplementary Information how latency in mosquitoes influences our basic results.

- Transmission of infection from vector to host and from host to vector is described by the parameters for the biting rate of each vector species, $b_1$ and $b_2$, the probability of successful transmission to and from a host species, $p_{ij}$, and preferences for each vector for one of the host species over the other, $\lambda_1$ and $\lambda_2$. The transmission rates from host to vector species are $\beta_{i3} = b_i\, p_{i3}$, $\beta_{i4} = b_i\, p_{i4}$, for $i \in \{1, 2\}$, and those from vector to host are $\beta_{i1} = b_1\, p_{i1}$, $\beta_{i2} = b_2\, p_{i2}$, for $i \in \{3, 4\}$. We assume transmission is host-frequency dependent [14] (i.e., the denominator in the expressions for the force of infection has the sum of all hosts), commonly used in mosquito-borne disease models reflecting satiation of the vectors.

- Vertical transmission in the vector species [15] is described by the parameter $q_i$, where $q_i = 0$ means that all offspring from an infectious vector is susceptible at birth, and where $q_i = 1$ means that all offspring of an infectious vector is infectious from birth.

- Horizontal transmission in the host species [16] is described by transmission rates $\beta_{33}$, $\beta_{44}$. These have been modelled as direct horizontal transmission only within the same species [16]. An alternative is to include transmission via carrion feeding [17], that is, where the virus is transmitted when susceptible birds feed on infected carcasses. The model can be extended by adding two new variables that collect all dead infected individuals of species. The inflow into these classes would be by death, and the outflow by being eaten or by natural decay. Here, however, we are only concerned with viral transmission directly between living species.

- The parameter $\lambda_i$ reflects a feeding preference of the vector species $N_i$ on one of the hosts relative to the other, $i \in \{1, 2\}$. The assumptions leading to the incorporation of $\lambda_i$ into the transmission terms in system (1) is explained in more detail later. For now, we assume that there are no feeding preferences, i.e., $\lambda_1 = \lambda_2 = 1$, and the mosquitoes have equal probabilities of biting each bird species.

Competition between species is assumed to affect the birth rate and (natural) death rate of the species involved. The factor $d_{ij}$, with $0 \leq d_{ij} \leq 1$, represents the proportion of competition from species $j$ that has an effect on the (natural) death rate of species $i$ (so the complementary fraction $1 - d_{ij}$ is the effect on the birth rate of species $i$). In other words, the actual death rate of species $N_1$ is in fact $\hat{\mu}_1 := \mu_1 + d_{11}c_{11}N_1 + d_{12}c_{12}N_2$ and the birth rate of species $N_1$ is $\hat{v}_1 := v_1 - (1 - d_{11})c_{11}N_1 - (1 - d_{12})c_{12}N_2$. Therefore, the factors $d_{ij}$ appear only in the

equations of the infected subsystem as they cancel at the level of the total population sizes where birth and natural death and all competition are accounted for. This holds both for intra and interspecific competition.

We want to study invasion and hence we take as our starting point the infection-free steady state of system (1) with all four species present at their respective infection-free equilibrium abundances, given by

$$
\begin{aligned}
N_1^* &= (c_{22}\mu_1 - c_{12}\mu_2 - c_{22}v_1 + c_{12}v_2)/(c_{12}c_{21} - c_{11}c_{22}) \\
N_2^* &= (c_{11}\mu_2 - c_{21}\mu_1 - c_{11}v_2 + c_{21}v_1)/(c_{12}c_{21} - c_{11}c_{22}) \\
N_3^* &= (c_{44}\mu_3 - c_{34}\mu_4 - c_{44}v_3 + c_{34}v_4)/(c_{34}c_{43} - c_{33}c_{44}) \\
N_4^* &= (c_{33}\mu_4 - c_{43}\mu_3 - c_{33}v_4 + c_{43}v_3)/(c_{34}c_{43} - c_{33}c_{44}).
\end{aligned}
$$

In order for these steady state abundances to be positive, the death rates must be lower than the birth rates for each species, and the competition terms must respect the following constraints: $c_{ii}\mu_j + c_{ji}v_i < c_{ji}\mu_i + c_{ii}v_j$ and $c_{jj}\mu_i + c_{ij}v_j < c_{ij}\mu_j + c_{jj}v_i$, with $\{i, j\} = \{1, 2\}$ for the vectors and $\{i, j\} = \{3, 4\}$ for the hosts. If we assume the birth rates to always be higher than the death rates, this steady state is stable in the absence of infection when the intraspecific competition is stronger than the interspecific competition [25].

The Jacobian in the infection-free steady state with all four species present for the above system is given by (see [13]):

$$
J = \begin{pmatrix} A & B \\ 0 & H \end{pmatrix}
$$

where

$$
A = \begin{pmatrix}
a_{11} & -c_{12}N_1 & 0 & 0 \\
-c_{21}N_2 & a_{22} & 0 & 0 \\
0 & 0 & a_{33} & -c_{34}N_3 \\
0 & 0 & -c_{43}N_4 & a_{44}
\end{pmatrix},
\qquad
B = \begin{pmatrix}
0 & 0 & 0 & 0 \\
0 & 0 & 0 & 0 \\
0 & 0 & -\alpha_3 & 0 \\
0 & 0 & 0 & -\alpha_4
\end{pmatrix}
$$

with $a_{11} = v_1 - \mu_1 - 2c_{11}N_1 - c_{12}N_2$, $a_{22} = v_2 - \mu_2 - c_{21}N_1 - 2c_{22}N_2$, $a_{33} = v_3 - \mu_3 - 2c_{33}N_3 - c_{34}N_4$, and $a_{44} = v_4 - \mu_4 - c_{43}N_3 - 2c_{44}N_4$, and

$$
H = \begin{pmatrix}
h_{11} & 0 & \dfrac{\beta_{13}\lambda_1 N_1}{\lambda_1 N_3 + N_4} & \dfrac{\beta_{14}N_1}{\lambda_1 N_3 + N_4} \\[2ex]
0 & h_{22} & \dfrac{\beta_{23}\lambda_2 N_2}{\lambda_2 N_3 + N_4} & \dfrac{\beta_{24}N_2}{\lambda_2 N_3 + N_4} \\[2ex]
\dfrac{\beta_{31}\lambda_1 N_3}{\lambda_1 N_3 + N_4} & \dfrac{\beta_{32}\lambda_2 N_3}{\lambda_2 N_3 + N_4} & h_{33} & 0 \\[2ex]
\dfrac{\beta_{41}N_4}{\lambda_1 N_3 + N_4} & \dfrac{\beta_{42}N_3}{\lambda_2 N_3 + N_4} & 0 & h_{44}
\end{pmatrix}
$$

with $h_{11} = q_1 - (c_{11}d_{11}N_1 + c_{12}d_{12}N_2)v_1 - \mu_1$, $h_{22} = q_2 - (c_{21}d_{21}N_1 + c_{22}d_{22}N_2)v_2 - \mu_2$, $h_{33} = \frac{\beta_{33}N_3}{N_3+N_4} - (c_{33}d_{33}N_3 + c_{34}d_{34}N_4)v_3 - \mu_3 - \alpha_3$, and $h_{44} = \frac{\beta_{44}N_4}{N_3+N_4} - (c_{43}d_{43}N_3 + c_{44}d_{44}N_4)v_4 - \mu_4 - \alpha_4$.

Here, matrix $A$ contains the ecological variables, matrix $H$ contains the epidemiological variables, and matrix $B$ reflects their interaction. The infection-free steady state is ecologically stable if all eigenvalues of $A$ have negative real parts and epidemiologically stable if all eigenvalues of $H$ have negative real parts. The matrix $H$ relates to the Next Generation Matrix that characterizes $R_0$ [13]. One can show that the infection-free steady state with all four species present is ecologically stable whenever it exists.

We can split matrix $H$ as $H = T + \Sigma$, where $T$ contains the epidemiological transmission terms and $\Sigma$ the epidemiological transitions. For our current system these are given by

$$
T = \begin{pmatrix}
q_1 v_1 & 0 & \dfrac{\beta_{13} \lambda_1 N_1}{\lambda_1 N_3 + N_4} & \dfrac{\beta_{14} N_1}{\lambda_1 N_3 + N_4} \\[2ex]
0 & q_2 v_2 & \dfrac{\beta_{23} \lambda_2 N_2}{\lambda_2 N_3 + N_4} & \dfrac{\beta_{24} N_2}{\lambda_2 N_3 + N_4} \\[2ex]
\dfrac{\beta_{31} \lambda_1 N_3}{\lambda_1 N_3 + N_4} & \dfrac{\beta_{32} \lambda_2 N_3}{\lambda_2 N_3 + N_4} & \dfrac{\beta_{33} N_3}{N_3 + N_4} & 0 \\[2ex]
\dfrac{\beta_{41} N_4}{\lambda_1 N_3 + N_4} & \dfrac{\beta_{42} N_3}{\lambda_2 N_3 + N_4} & 0 & \dfrac{\beta_{44} N_4}{N_3 + N_4}
\end{pmatrix}
$$

and

$$
\Sigma = \begin{pmatrix}
\sigma_{11} & 0 & 0 & 0 \\
0 & \sigma_{22} & 0 & 0 \\
0 & 0 & \sigma_{33} & 0 \\
0 & 0 & 0 & \sigma_{44}
\end{pmatrix}
$$

with $\sigma_{11} = q_1(-c_{11}(1 - d_{11})N_1 - c_{12}(1 - d_{12})N_2) - c_{11} d_{11} N_1 - c_{12} d_{12} N_2 - \mu_1$, $\sigma_{22} = q_2(-c_{21}(1 - d_{21})N_1 - c_{22}(1 - d_{22})N_2)N_2) - c_{21} d_{21} N_1 - c_{22} d_{22} N_2 - \mu_2$, $\sigma_{33} = -c_{33} d_{33} N_3 - c_{34} d_{34} N_4 - \alpha_3 - \mu_3$, and $\sigma_{44} = -c_{43} d_{43} N_3 - c_{44} d_{44} N_4 - \alpha_4 - \mu_4$.

The element at position $(i, i)$ in $\Sigma$ can be interpreted as the rate of leaving compartment $i$. In our current set-up all other elements of $\Sigma$ are zero given our assumptions. The $(i, j)$ entry of $T$ is the rate at which infected individuals of type $j$ produce new infections of type $i$. Then, multiplying the two matrices, $K = -T \Sigma^{-1}$, results in a matrix where the entry $(i, j)$ has the interpretation of the expected number of new infections of type $i$ produced by an individual that started infected life as type $j$. This matrix is called the Next Generation Matrix and is given by

$$
K = \begin{pmatrix}
k_{11} & 0 & k_{13} & k_{14} \\
0 & k_{22} & k_{23} & k_{24} \\
k_{31} & k_{32} & k_{33} & 0 \\
k_{41} & k_{42} & 0 & k_{44}
\end{pmatrix}
$$

with

$$k_{11} = \frac{q_1 v_1}{c_{11}N_1(d_{11} + q_1 - d_{11}q_1) + c_{12}N_2(d_{12} + q_1 - d_{12}q_1) + \mu_1}$$

$$k_{13} = \frac{\beta_{13}\lambda_1 N_1}{(\lambda_1 N_3 + N_4)(c_{33}d_{33}N_3 v_3 + c_{34}d_{34}N_4 v_3 + \mu_3 + \alpha_3)}$$

$$k_{14} = \frac{\beta_{14} N_1}{(\lambda_1 N_3 + N_4)(c_{43}d_{43}N_3 v_4 + c_{44}d_{44}N_4 v_4 + \mu_4 + \alpha_4)}$$

$$k_{22} = \frac{q_2 v_2}{c_{21}N_1(d_{21} + q_2 - d_{21}q_2) + c_{22}N_2(d_{22} + q_2 - d_{22}q_2) + \mu_2}$$

$$k_{23} = \frac{\beta_{23}\lambda_2 N_2}{(\lambda_2 N_3 + N_4)(c_{33}d_{33}N_3 v_3 + c_{34}d_{34}N_4 v_3 + \mu_3 + \alpha_3)}$$

$$k_{24} = \frac{\beta_{24} N_2}{(\lambda_2 N_3 + N_4)(c_{43}d_{43}N_3 v_4 + c_{44}d_{44}N_4 v_4 + \mu_4 + \alpha_4)}$$

$$k_{31} = \frac{\beta_{31}\lambda_1 N_3}{(\lambda_1 N_3 + N_4)(c_{11}N_1(d_{11} + q_1 - d_{11}q_1) + c_{12}N_2(d_{12} + q_1 - d_{12}q_1) + \mu_1)}$$

$$k_{32} = \frac{\beta_{32}\lambda_2 N_3}{(\lambda_2 N_3 + N_4)(c_{21}N_1(d_{21} + q_2 - d_{21}q_2) + c_{22}N_2(d_{22} + q_2 - d_{22}q_2) + \mu_2)}$$

$$k_{33} = \frac{\beta_{33} N_3}{(N_3 + N_4)(c_{33}d_{33}N_3 v_3 + c_{34}d_{34}N_4 v_3 + \mu_3 + \alpha_3)}$$

$$k_{41} = \frac{\beta_{41} N_4}{(\lambda_1 N_3 + N_4)(c_{11}N_1(d_{11} + q_1 - d_{11}q_1) + c_{12}N_2(d_{12} + q_1 - d_{12}q_1) + \mu_1)}$$

$$k_{42} = \frac{\beta_{42} N_4}{(\lambda_2 N_3 + N_4)(c_{21}N_1(d_{21} + q_2 - d_{21}q_2) + c_{22}N_2(d_{22} + q_2 - d_{22}q_2) + \mu_2)}$$

$$k_{44} = \frac{\beta_{44} N_4}{(N_3 + N_4)(c_{43}d_{43}N_3 v_4 + c_{44}d_{44}N_4 v_4 + \mu_4 + \alpha_4)}$$

The basic reproduction number $R_0$ is the dominant eigenvalue of $K$ [26] and measures the epidemiological stability of the infection-free steady state, i.e. when $N_i = N_i^*$ and $I_i = 0$, $i \in \{1, 2, 3, 4\}$. If $R_0 > 1$ the infection-free steady state is unstable and the pathogen successfully invades in the community described by our model. Since this four dimensional matrix has elements along the diagonal, it is not possible to derive an expression for $R_0$ analytically. We can instead, explore it numerically using the maximum real eigenvalue provided by the R [27] function `eigen`, applied to $K$.

We study numerically, using WNV infection to inform the parameter values, how $R_0$ depends on the ecological interactions in our system.

## Multi-vector multi-host system with vector feeding preferences

We are also interested in studying how vector feeding preferences in a multi-vector multi-host competitive environment may affect infection dynamics. These preferences are modelled in similar fashion as in [28], yet extended here to a system with multiple vectors. Suppose first there is only one individual mosquito (species 1) and two types of hosts (3 and 4). Host species 3 has $N_3$ members, host species 4 has $N_4$ members. So, in total there are $N_3 + N_4$ host individuals in the area of interest. It is assumed that the system is closed and that the mosquito does not bite any other animals. A single vector individual bites hosts $b_1$ times each day, and we wish to know how these bites are to be divided over the two host species. Let $w_1$ be the probability for one randomly selected host individual to get a bite (on a given day) from a vector individual of species 1, if there is no biting preference. Then $w_1 = b_1/(N_3 + N_4)$ if biting is with

replacement (in other words, all bites by the mosquito are independent events). If biting is random, i.e., when there is no preference at all for either of the two host species, then species 3 will receive $w_1 N_3$ of the $b_1$ bites and species 4 will receive $w_1 N_4$ of the $b_1$ bites. In total this gives

$$w_1 N_3 + w_1 N_4 = \frac{b_1 N_3}{N_3 + N_4} + \frac{b_1 N_4}{N_3 + N_4} = b_1 \qquad (2)$$

Now assume that the mosquito has a preference for one of the host species. We denote by $\lambda_1$ the feeding preference as the 'weight' that the mosquito attaches to one of the host species, compared to the other. We define that relative to host species 3, without loss of generality. The question is how this could be accommodated and how we would then express the number of bites that go to species 3 and to species 4. The preference $\lambda_1$ is an abstract concept, but can thought of in the following way. Instead of $N_3$ individuals of type 3 and $N_4$ individuals of type 4, we act as if there are $\lambda_1 N_3$ of type 3 (from the point of view of the vector) and $N_4$ of type 4. This artificial population now gives us the option to define the probability $r_1$, that a randomly selected individual gets bitten when there is preference, as $r_1 = b_1/(\lambda_1 N_3 + N_4)$. Note that $\lambda_1 = 1$ implies that $r_1 = w_1$. In that scenario, species 3 will receive $p_1 \lambda_1 N_3$ of the $b_1$ bites, and species 4 will receive $r_1 N_4$ of the bites. This means that

$$r_1 \lambda_1 N_3 + r_1 N_4 = \frac{b_1 \lambda_1 N_3}{\lambda_1 N_3 + N_4} + \frac{b_1 N_4}{\lambda_1 N_3 + N_4} = b_1 \qquad (3)$$

Now suppose we have two vector species. The reasoning is the same as the one derived above, but now with two biting rates $b_1$ and $b_2$ and with two preferences $\lambda_1$ and $\lambda_2$, which may be different. After all, it is the maximum bites per day that are set for each vector species, and the way these are divided over the hosts is then determined by the host population sizes and the preference parameters.

If $\lambda_1 < 1$, vector species 1 has a preferences for host species 4 (with the extreme case being $\lambda_1 = 0$ where this vector only feeds on host species 4). If $\lambda_1 = 1$, vector species 1 has no preference and bites randomly according to the relative abundances of the two host species. Finally, for $\lambda_1 > 1$, vector species 1 preferentially bites host species 3. The same interpretation holds for $\lambda_2$, the feeding preference of vector species $N_2$ towards one of the hosts.

## Including non-competent vectors and hosts

As stated before, the dynamics of an infection in a community may also be affected by species that do not contribute directly to the transmission events [11]. Non-competent species, whether susceptible or not to a given pathogen, can interact with the competent vectors and hosts of that pathogen affecting infection dynamics and outbreak risk. We now consider some extensions to the model by introducing non-competent species to the system.

It is important to first clarify the different types of species in a community from the point of view of a given pathogen. By pathogen non-competence we mean the inability to transmit the pathogen. This can be because the species is not even susceptible but can also be because the infection in individuals of the species does not lead to the type or quantity of pathogen multiplication to allow successful transmission to other individuals or species. The former group of species are non-hosts for the pathogen and comprises most species in a community at all trophic levels for any specific pathogen. The latter are the so-called dead-end host species. Vector transmission leads to additional types of species because in principle a vector species could take blood meals from a species that is pathogen non-competent, both non-host species suitable for completing the vector life cycle and dead-end host species, such as humans. Most

species, at all trophic levels, will be pathogen non-competent as well as unsuitable from the point of view of the vector. Also, in the group of potential vector species one may recognise both competent and non-competent species or individuals, for example in the case of genetically modified mosquitoes or mosquitoes treated by some means of control. All types of species can, through their direct and indirect interactions, ecologically and epidemiologically, affect vector and pathogen dynamics.

Here, we restrict the analysis to non-competent vectors (species or individuals) and dead-end host species. In mathematical terms, non-host species that attract mosquito bites are indistinguishable from dead-end hosts. In both cases, these non-competent vectors and hosts are infected with WNV, but the infectious mosquito bites are wasted as transmission opportunities. The analysis below therefore relates both to dead-end hosts and other non-competent species, as long as they are suitable for the life cycle of the vector species.

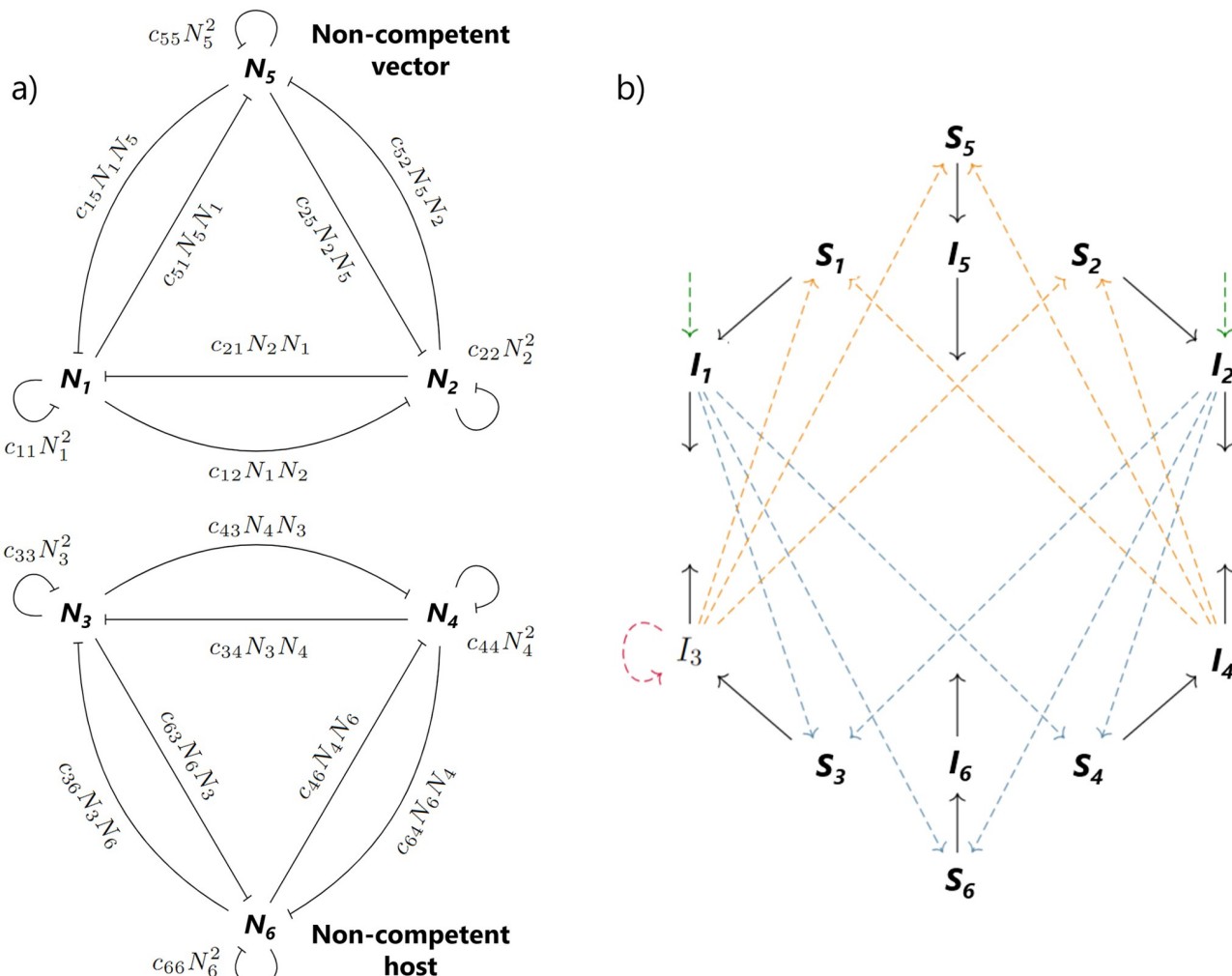

**Fig 2. Diagrams of the ecological (a) and epidemiological (b) components of the model with non-competent species.** Note how the non-competent vectors, $N_5$, and non-competent hosts, $N_6$, can become infected, but do not further transmit WNV. T-bars: competition, black arrows: infection related transitions, dashed arrows: transmission routes (green: vertical, yellow: host-to-vector, blue: vector-to-host, red: horizontal).

Studying non-competent vectors can be relevant when it concerns a species of vector related to a competent vector species in the sense that it occupies the same ecological niche and can be expected to exert an influence via competition in the environment, for example at the larval stage. Impact of non-competent vectors on pathogen emergence is also relevant to study when considering interventions with modified or treated vector individuals that will disrupt a population of their wild-type vector species members. Their impact on pathogen emergence should be studied when considering interventions with additional vectors, such as the introduction of genetically modified mosquitoes [29] or with *Wolbachia*-induced incompatibility strategies [30]. These non-competent vectors, we argue, can still indirectly impact pathogen dynamics in a community through competition with the other competent mosquitoes. Likewise, some host species do not develop levels of viremia sufficient for successful transmission to a vector. For WNV, it is believed that the Mourning Dove (*Zenaida macroura*) is responsible for a dilution effect [31], so we use this species as an example. These dead-end host species, despite not contributing directly to transmission, are still present in system and will interact with the remaining competent species in ways that could even lead to an amplification of the disease risk. Non-competent hosts behave differently from non-competent vectors in the sense that they can have an impact on infection dynamics even if they do not compete with competent hosts. We argue that the mere presence of non-hosts can have a significant impact on disease invasion.

To answer these questions, we extend the previous model to include a non-competent vector $N_5$ and a non-competent host $N_6$ (Fig 2), given now by

$$
\begin{aligned}
N_1' &= v_1 N_1 - (c_{11}N_1 + c_{12}N_2 + c_{15}N_5 + \mu_1)N_1 \\
N_2' &= v_2 N_2 - (c_{21}N_1 + c_{22}N_2 + c_{25}N_5 + \mu_2)N_2 \\
N_3' &= v_3 N_3 - (c_{33}N_3 + c_{34}N_4 + c_{36}N_6 + \mu_3)N_3 - \alpha_3 I_3 \\
N_4' &= v_4 N_4 - (c_{43}N_3 + c_{44}N_4 + c_{46}N_6 + \mu_4)N_4 - \alpha_4 I_4 \\
N_5' &= v_5 N_5 - (c_{51}N_1 + c_{52}N_2 + c_{55}N_5 + \mu_5)N_5 \\
N_6' &= v_6 N_6 - (c_{63}N_3 + c_{64}N_4 + c_{66}N_6 + \mu_6)N_6 - \alpha_6 I_6 \\
I_1' &= q_1(v_1 - (1-d_{11})c_{11}N_1 - (1-d_{12})c_{12}N_2 - (1-d_{15})c_{15}N_5)I_1 \\
&\quad + \frac{S_1(\beta_{13}I_3 + \beta_{14}I_4)}{N_3 + N_4 + N_6} - (c_{11}d_{11}N_1 + c_{12}d_{12}N_2 + c_{15}d_{15}N_5 + \mu_1)I_1 \\
I_2' &= q_2(v_2 - (1-d_{21})c_{21}N_1 - (1-d_{22})c_{22}N_2 - (1-d_{25})c_{25}N_5)I_2 \\
&\quad + \frac{S_2(\beta_{23}I_3 + \beta_{24}I_4)}{N_3 + N_4 + N_6} - (c_{21}d_{21}N_1 + c_{22}d_{22}N_2 + c_{25}d_{25}N_5 + \mu_2)I_2 \\
I_3' &= \frac{S_3(\beta_{31}I_1 + \beta_{32}I_2 + \beta_{33}I_3)}{N_3 + N_4 + N_6} - (c_{33}d_{33}N_3 + c_{34}d_{34}N_4 + c_{36}d_{36}N_6 + \mu_3 + \alpha_3)I_3 \\
I_4' &= \frac{S_4(\beta_{41}I_1 + \beta_{42}I_2 + \beta_{44}I_4)}{N_3 + N_4 + N_6} - (c_{43}d_{43}N_3 + c_{44}d_{44}N_4 + c_{46}d_{46}N_6 + \mu_4 + \alpha_4)I_4 \\
I_5' &= q_5(v_5 - (1-d_{51})c_{51}N_1 - (1-d_{52})c_{52}N_2 - (1-d_{55})c_{55}N_5)I_5 \\
&\quad + \frac{S_5(\beta_{53}I_3 + \beta_{54}I_4)}{N_3 + N_4 + N_6} - (c_{51}d_{51}N_1 + c_{52}d_{52}N_2 + c_{55}d_{55}N_5 + \mu_5)I_5 \\
I_6' &= \frac{S_6(\beta_{61}I_1 + \beta_{62}I_2 + \beta_{66}I_6)}{N_3 + N_4 + N_6} - (c_{63}d_{63}N_3 + c_{64}d_{64}N_4 + c_{66}d_{66}N_6 + \mu_6 + \alpha_6)I_6
\end{aligned}
\tag{4}
$$

with the new parameters listed in Table 2. Again, $\beta_{ij} = p_{ij} b_j$ for the relevant values of $i$ and $j$. For simplicity, we ignore any vector feeding preferences in this section and assume equivalent parameter values as used before.

**Table 2. Additional parameters regarding simulations with non-competent species.**

| Parameter | Description | Estimate |
|---|---|---|
| $v_5$ | Non-competent vector birth rate | 0.537 |
| $v_6$ | Non-competent host birth rate | 0.5 |
| $\mu_5$ | Non-competent vector death rate | 0.08 |
| $\mu_6$ | Non-competent host death rate | $2.7 \times 10^{-4}$ |
| $\alpha_6$ | Additional disease-related death rate in the non-competent host | 0.2 |
| $p_{61}$ | Transmission probability *C. pipiens* to non-competent host | 0.80 |
| $p_{62}$ | Transmission probability *C. molestus* to non-competent host | 0.80 |
| $b_1$ | Biting rate of *C. pipiens* on non-competent host | 0.4 |
| $b_2$ | Biting rate of *C. molestus* on non-competent host | 0.4 |

There are several solutions for this system. We are only interested in an infection-free steady state where all species are present, and for this we study the following:

$$
\begin{aligned}
N_1^* &= (c_{22}c_{55}(\mu_1 - v_1) + c_{25}c_{52}(v_1 - \mu_1) \\
&\quad + (c_{15}c_{52} - c_{12}c_{55})(\mu_2 - v_2) + c_{12}c_{25}(\mu_5 - v_5) + c_{15}c_{22}(v_5 - \mu_5)) \\
&\quad /(c_{15}c_{22}c_{51} - c_{12}c_{25}c_{51} - c_{15}c_{21}c_{52} + c_{11}c_{25}c_{52} + c_{12}c_{21}c_{55} - c_{11}c_{22}c_{55}) \\
N_2^* &= (c_{21}c_{55}(\mu_1 - v_1) + c_{25}c_{51}(v_1 - \mu_1) \\
&\quad + (c_{15}c_{51} - c_{11}c_{55})(\mu_2 - v_2) + c_{11}c_{25}(\mu_5 - v_5) + c_{15}c_{21}(v_5 - \mu_5)) \\
&\quad /(-c_{15}c_{22}c_{51} + c_{12}c_{25}c_{51} + c_{15}c_{21}c_{52} - c_{11}c_{25}c_{52} - c_{12}c_{21}c_{55} + c_{11}c_{22}c_{55}) \\
N_3^* &= (c_{44}c_{66}(\mu_3 - v_3) + c_{46}c_{64}(v_3 - \mu_3) \\
&\quad + (c_{36}c_{64} - c_{34}c_{66})(\mu_4 - v_4) + c_{34}c_{46}(\mu_6 - v_6) + c_{36}c_{44}(v_6 - \mu_6)) \\
&\quad /(c_{36}c_{44}c_{63} - c_{34}c_{46}c_{63} - c_{36}c_{43}c_{64} + c_{33}c_{46}c_{64} + c_{34}c_{43}c_{66} - c_{33}c_{44}c_{66}) \\
N_4^* &= (c_{43}c_{66}(\mu_3 - v_3) + c_{46}c_{63}(v_3 - \mu_3) \\
&\quad + (c_{36}c_{63} - c_{33}c_{66})(\mu_4 - v_4) + c_{33}c_{46}(\mu_6 - v_6) + c_{36}c_{43}(v_6 - \mu_6)) \\
&\quad /(-c_{36}c_{44}c_{63} + c_{34}c_{46}c_{63} + c_{36}c_{43}c_{64} - c_{33}c_{46}c_{64} - c_{34}c_{43}c_{66} + c_{33}c_{44}c_{66}) \\
N_5^* &= (c_{21}c_{52}(\mu_1 - v_1) + c_{22}c_{51}(v_1 - \mu_1) \\
&\quad + (c_{12}c_{51} - c_{11}c_{52})(\mu_2 - v_2) + c_{11}c_{22}(\mu_5 - v_5) + c_{12}c_{21}(v_5 - \mu_5)) \\
&\quad /(c_{15}c_{22}c_{51} - c_{12}c_{25}c_{51} - c_{15}c_{21}c_{52} + c_{11}c_{25}c_{52} + c_{12}c_{21}c_{55} - c_{11}c_{22}c_{55}) \\
N_6^* &= (c_{43}c_{64}(\mu_3 - v_3) + c_{44}c_{63}(v_3 - \mu_3) \\
&\quad + (c_{34}c_{63} - c_{33}c_{64})(\mu_4 - v_4) + c_{33}c_{44}(\mu_6 - v_6) + c_{34}c_{43}(v_6 - \mu_6)) \\
&\quad /(c_{36}c_{44}c_{63} - c_{34}c_{46}c_{63} - c_{36}c_{43}c_{64} + c_{33}c_{46}c_{64} + c_{34}c_{43}c_{66} - c_{33}c_{44}c_{66}).
\end{aligned}
$$

As we assume that vectors only compete with vectors, and hosts only compete with hosts, our system can be separated into two systems of three-species competition. To have feasible (non-negative) solutions for a three-species competition system the intraspecific competition must again be stronger than the interspecific competition [32].

## Results

### Numerical example: Impact of competition on WNV invasion risk

We calculated the invasion risk of WNV into a community consisting of competitive vectors (*Culex pipiens* and *Culex molestus*) and competitive hosts (American crow and House finch).

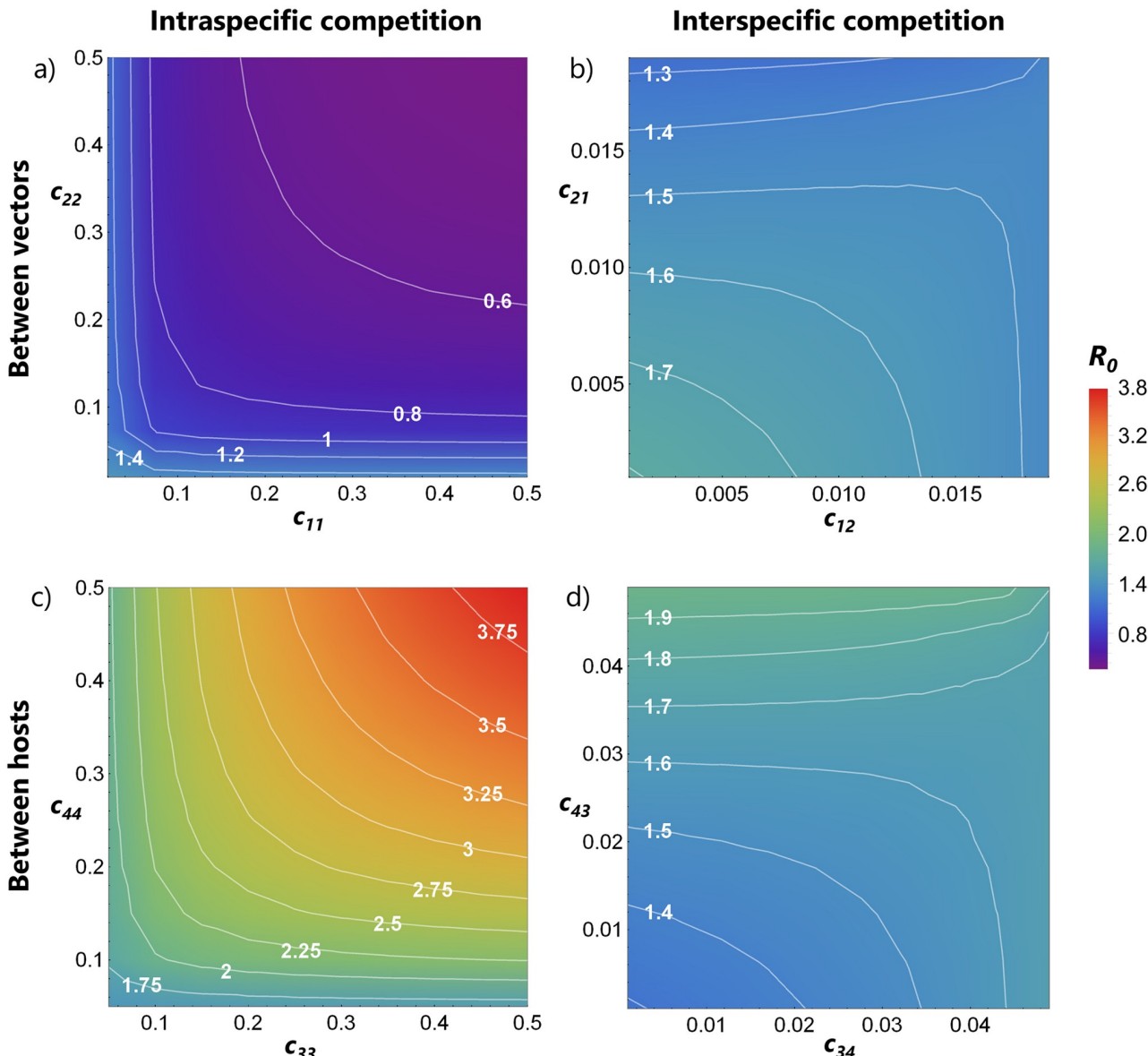

**Fig 3. $R_0$ values based on intraspecific (a and c) and interspecific competition (b and d) between vectors (a and b) and between hosts (c and d).** All the remaining parameters are fixed at their point estimates shown in Table 1.

The direct effect of the competition strengths on $R_0$ was investigated by looking at a range of values for the parameters $c_{ij}$, with $i \neq j$. The competition terms $c_{ii}$ are varied over the range $[c_{ij} + 0.01, 0.5]$ and $c_{ij}$ over $[0, 0.02]$ for vectors and over $[0, 0.05]$ for hosts. In this way, we can explore a large part of competition parameter space, while still respecting the conditions that guarantee that all species are present. Fig 3 shows the effect of competition between the different mosquito species and between the bird species on $R_0$, assuming all other parameter values are fixed. Competition among hosts has a much wider range of possible $R_0$ outcomes compared with competition between vectors (so much so that different scales need to be used to see the patterns from vector competition). The host species differ slightly in terms of competence and how their death rates are increased by the disease (see Table 1), yet the resulting

invasion risk patterns based on the competition parameters are quite symmetrical. Intraspecific competition among hosts results in the widest range of $R_0$ values (Fig 3c). In all cases, apart from intraspecific competition between vectors, $R_0 > 1$. The case where intraspecific vector competition results in $R_0 < 1$ (Fig 3a) is interesting, as it may be observed in years or regions with low precipitation, resulting in reduced breeding sites for mosquitoes, and hence more competition for resources during their early development stages [5]. The impact of all other parameters on $R_0$ is summarized in our elasticity analysis in the Supporting Information.

One way to look at pathogen invasion risk upon introduction in communities is by looking at the constituent species abundances. If there is an increase in a key transmitting species, a higher risk of an outbreak could be expected. In networks with several interacting vectors and hosts, this assessment may be made more difficult. It is often helpful to regard the vector-to-host ratio $v/h$ in a system, i.e., the vector population size divided by the host population size, or in other words: the mean number of vectors in the system per host individual. In the basic Ross-Macdonald model, with one vector species and one host species and pure vector transmission, $R_0$ is an increasing function of $v/h$ [33]. In systems with more species, the relation becomes more complicated, both analytically and because of the large number of different ways in which to characterize $v/h$. Here, we have defined it as $v/h = (N_1 + N_2)/(N_3 + N_4)$, which we refer to as the 'overall $v/h$-ratio'. We note, however, that many different and meaningful $v/h$-ratios could be calculated, for example the abundance of only one vector species divided by total host species' abundance, or divided by the abundance of only one host species. We chose the overall ratio above as it includes the contribution of every species, that is, the total vector abundance divided by the total host abundance. For our coupled system with two vector species and two host species, we are not able to show analytically that $R_0$ increases with the overall vector-to-host ratio. However, we can give an indication through the use of simulations by randomly changing the values of the competition parameters. Each combination of parameter values, and the resulting population abundances, produces its own (overall) vector-to-host ratio in the definition given before and, consequently, is associated with a certain $R_0$ value.

Several life-history parameters can determine the observed species abundances, but we focus on how competition relates to the overall $v/h$-ratio. For example, if there are fewer non-infected birds as a result of competition, the chance of a susceptible mosquito biting an infected bird can increase. In Fig 4 we show how, even in a multi-vector multi-host environment, there is still a general positive relationship between the overall $v/h$-ratio and $R_0$. We consider two competition scenarios. In the first, Fig 4a, the intra and interspecific competition parameters act by affecting mostly the birth rates (expressed by choosing $d_{ii} = 0.1$ and $d_{ij} = 0.05$), as we have assumed for the rest of the study. In the second, Fig 4b, competition acts mostly on the death rates (expressed by choosing $d_{ii} = 0.9$ and $d_{ij} = 0.85$). In both cases, we vary the competition coefficients of vectors, while keeping the host competition fixed, and vice-versa. For the vectors, $c_{ii}$ are uniformly sampled from the interval [0.02, 0.5] and $c_{ij}$ from [0, 0.019], and for hosts $c_{ii}$ are sampled from [0.05, 0.5] and $c_{ij}$ from [0, 0.049]. Each combination of competition coefficients results in a specific $v/h$-ratio, and a respective $R_0$. We performed 1000 iterations for each scenario, with all other parameters fixed to the values shown in Table 1. The range of possible $R_0$ values is the widest when varying competition between hosts, and when it acts more strongly the birth rates rather than the death rates. The latter, if high, may prevent the invasion of WNV (Fig 4b), potentially by leading to earlier death of individuals through competition before they become infectious or before there are transmission opportunities. The way competition affects the overall $v/h$-ratio is explored in more detail in the Supporting Information (see S2 Fig in S1 File).

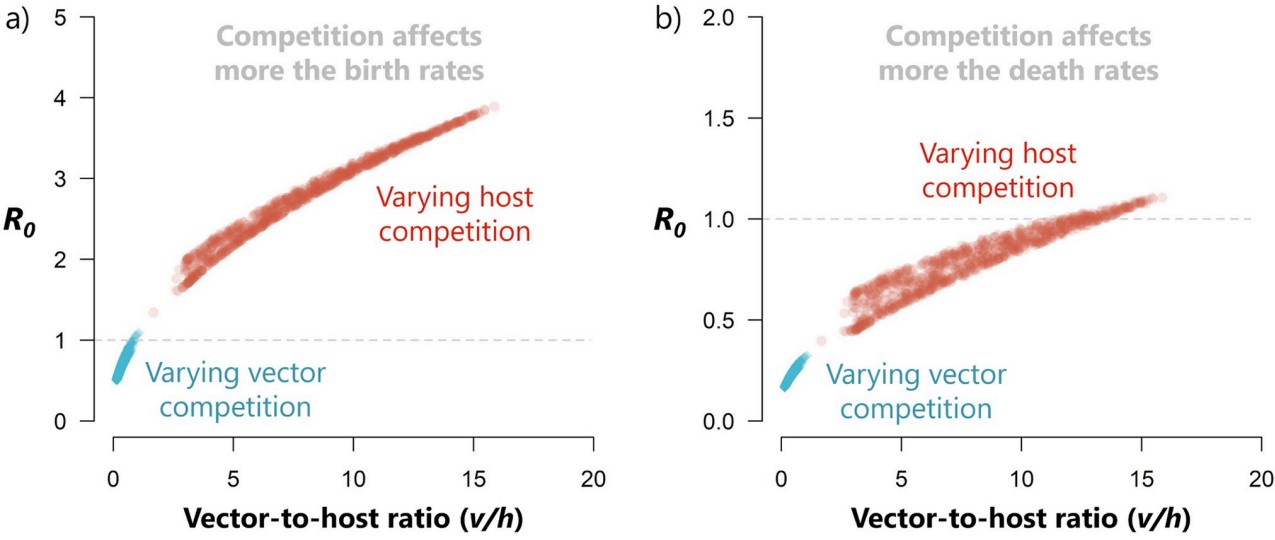

**Fig 4. Relationship between the vector-to-host ratio, $v/h$, and $R_0$.** The different values for $v/h$ are obtained by varying only the competition coefficients in vectors (blue diamonds) and in hosts (red circles). Two scenarios are considered: a) most competition acts on the birth rates of the affected species ($d_{ii} = 0.1$, $d_{ij} = 0.05$) or b) on the death rates ($d_{ii} = 0.9$, $d_{ij} = 0.85$).

### Vector feeding preferences under competitive conditions

In the scenario where the two mosquito species can show a preference towards one of the host species, we can imagine that differences in feeding preference only have potential impact on infection when the bird species involved differ in aspects relevant to the eco-epidemiological dynamics. Fig 5 shows the predicted basic reproduction number for a range of vector feeding preferences: in the top left corner vector $N_1$ prefers $N_4$ and $N_2$ prefers $N_3$; in the top right corner both vectors prefer to feed on $N_3$; in the bottom left both prefer $N_4$; and in the bottom right $N_1$ prefers $N_3$ and $N_2$ prefers $N_4$. Even very small differences in the bird species (see Table 1) in the presence of mosquito feeding preferences lead to very different ranges for $R_0$. More so, we see how the feeding preferences are made even more relevant when each host is under high competitive stress (Fig 5b and 5c), reflected by wider ranges for $R_0$. One way of

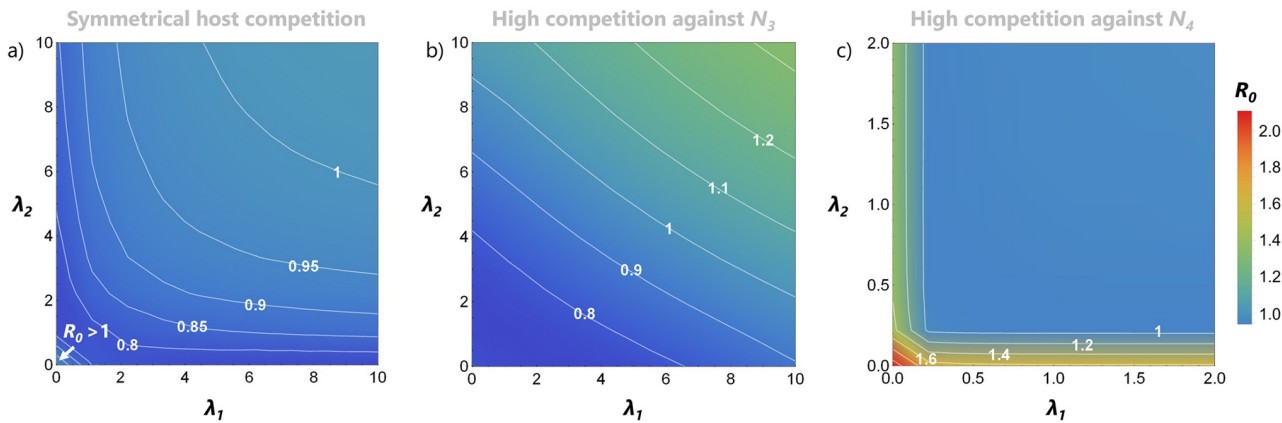

**Fig 5. Vector feeding preferences matter more under a stronger competitive burden.** (a-c) $R_0$ values depending on different mosquito feeding preferences, $\lambda_1$ and $\lambda_2$, under three scenarios of host competition: a) $c_{33} = c_{44}$, $c_{34} = c_{43}$; b) $2c_{33}$, $1.5c_{34}$; c) $1.5c_{43}$, $2c_{44}$ (with all other parameters fixed at the values in Table 1).

looking at this is: if a host is under strong competition (either from itself or from another species) then it is beneficial for invasion success if vectors would have preference for the host that is suffering the most from competition. That is, $R_0$ can be maximized when the mosquitoes feed on the least abundant birds, reflecting results in [34].

### Competitive pressures from non-competent vectors and hosts

We investigate then system (2) assuming that the non-competent species can have the same strengths of intra and interspecific competition as the competent species (i.e., we explore equivalent ranges for the competition coefficients as those used in the previous sections). We vary the competition coefficients $c_{ij}$ over the range [0, 0.02[ for vectors and over the range [0, 0.05[ for hosts. The presence of a non-competent host that does not compete, or competes very little, with the competent hosts can lead to a dilution effect. This results in lower values of $R_0$ as shown in Fig 6 (see regions where $c_{15}$, $c_{25}$, $c_{36}$, and $c_{46}$ are low). Note the regions delimited by the dashed line where $R_0 < 1$ which were not present when looking at interspecific competition in the model with only the competent species (see Fig 3b and 3d). This is because some mosquitoes are biting individuals of species that do not contribute to transmission.

However, the dilution effect from non-competent vectors is countered when they suffer from strong competition from the competent vectors (Fig 6a and 6b), and when non-competent hosts strongly compete against the competent hosts (Fig 6c and 6d). This can again be understood by thinking how the competition affects the vector-to-host ratio of competent species and consequently $R_0$. Strong competition against the most competent vectors decreases the overall $v/h$-ratio ($R_0 < 1$) when $c_{15}$ and $c_{25}$ is high, and strong competition against the most competent hosts increases the overall $v/h$-ratio ($R_0 > 1$) when $c_{46}$ is high. Therefore, to fully understand if non-competent species will facilitate or prevent disease invasion in a community it is important to look not only at how non-competent species affect biting rates but also at how they interact with the competent species.

## Discussion

The emergence of a pathogen in a community is dependent on the complex interplay of interactions between its species at all trophic levels. In this work, we used a mathematical model for a simplified community to explore invasion of a vector-borne disease into a multi-vector and multi-species setting. In this setting, the species interact both epidemiologically (through pathogen transmission) and ecologically (through competition for resources). Several potential additional transmission routes of infection were explored, such as horizontal transmission between birds of the same species and vertical transmission between the mosquitoes.

In this study, we were invested in studying how interactions, particularly competition, can influence the risk of disease emergence. Competition is an umbrella term and could lead to imprecise interpretations. Here we refer to competition for resources in the general sense and we make no comparisons of fitness differences [35]. The impact of competition between hosts on arbovirus invasion has been addressed (see for example [34]), but not, to the best of our knowledge, explicit competition between vectors, even though it may play a significant role in community ecology [36]. From our numerical analysis, we saw how competition at the vector level negatively affects $R_0$ by reducing the number of available mosquitoes that contribute to transmission of the virus in the system. On the other hand, between-host competition has an opposite impact, increasing $R_0$. These opposing effects will collectively affect invasion risk, depending on the strengths of the competition. This is in agreement with earlier modeling work on malaria transmission, which indicated that both increased mosquito competition and an increase in dead-end hosts can contribute to lower invasion risks [37].

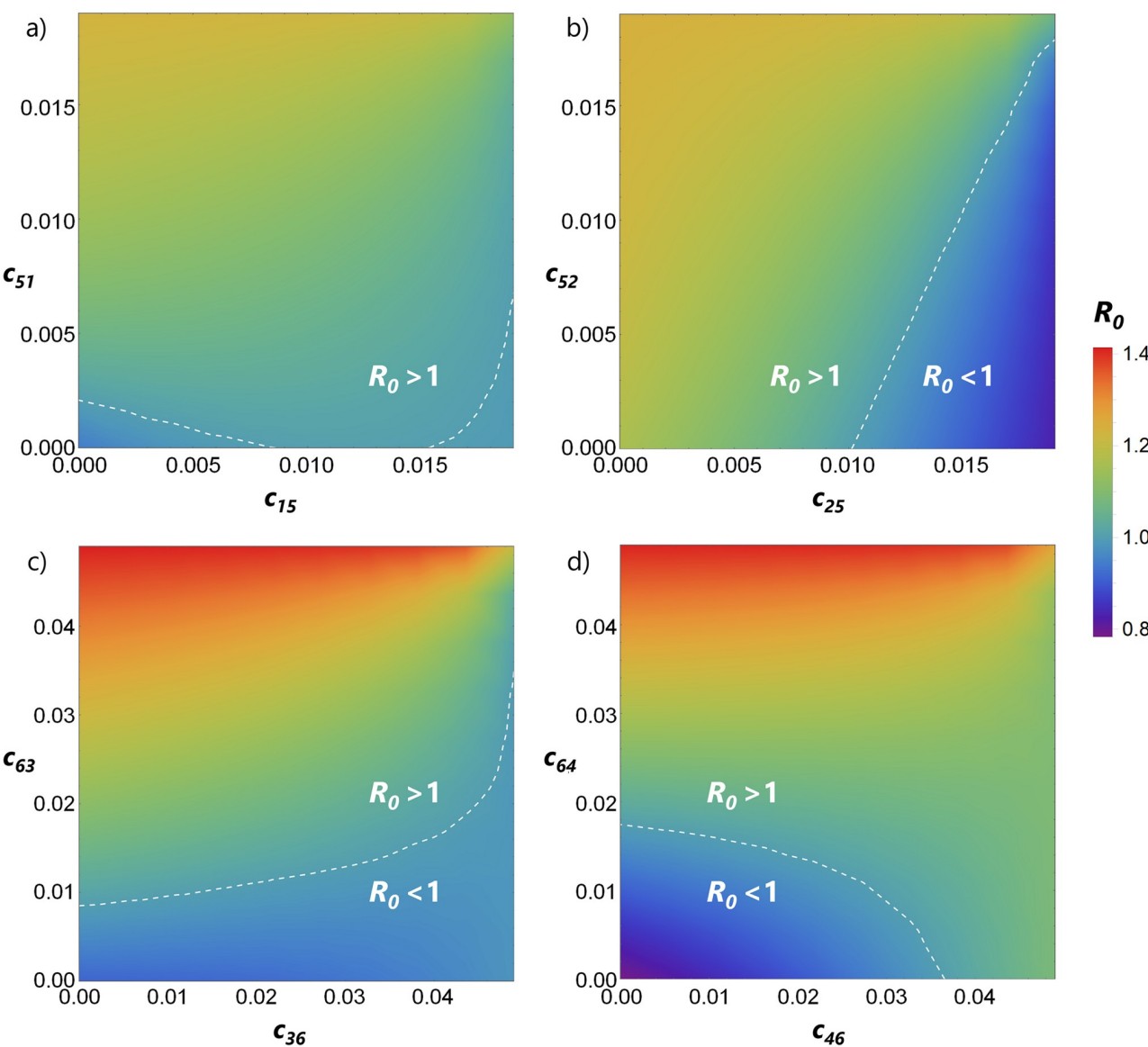

**Fig 6. Non-competent species have a general dilution effect, but the impact of their competitive pressures on $R_0$ differ for vectors and for hosts.** (a) and (b) interspecific competition with non-competent vector $N_5$, (c) and (d) interspecific competition with non-competent host $N_6$. In all cases, $R_0$ is maximized when the competition strength against the non-competent vectors or non-competent hosts is the highest. All other parameters are fixed to the estimates in Tables 1 and 2.

In the case of one vector species and one host species, $R_0$ is a linear function of the (uniquely defined) vector-to-host ratio. The way competition affects the vector-to-host ratio and consequently the transmission dynamics in a community has scarcely been explored, with a previous study indicating that competition between hosts is likely to result in an increase in the vector-to-host ratio [12]. In the case of more vector and host species, the relation becomes less straightforward in two ways. Firstly, there is no single vector-to-host ratio but rather a set of such numbers, depending on what vector and what host species are considered. An obvious choice would be what we call the 'overall vector-to-host ratio', calculated as the total abundance of all vector species divided by the total abundance of all host species. Secondly, the vector-to-host ratio, in any definition, is no longer a separate factor in $R_0$ and the influence of this

ratio on $R_0$ can no longer be determined directly. Contrary to the single vector, single host case, we observed a non-linear increase in $R_0$ as the overall vector-to-host ratio increases. We advise some caution when interpreting these results [38]. If indeed a high vector-to-host ratio leads to a high $R_0$, then regions with almost no hosts and many vectors would be at the highest risk. However, since the vectors are dependent on the hosts for blood-feeding, in the long-term these regions would not be sustained. This is nonetheless a fair assumption here since we consider that the host population is large and hence not limiting to the vector feeding.

We assumed a competitive burden at the same stage for both vectors and hosts. Mosquitoes are more likely to share a niche, and hence, compete the most (for space and nutrients) during the early larval stages [5], while competition in birds (for territory and food) can be expected to be stronger when they are adults [7]. Then, in future studies, it would be fair to assume low values for the amount of intra and interspecific competition acting at birth (given by $d_{ii}$ and $d_{ij}$ respectively). Furthermore, since intraspecific competition reflects the carrying capacity of a given niche to a certain species, the factors $d_{ii}$ and $d_{ij}$ may differ significantly. One could expect some species to suffer more from intraspecific competition at birth, and then as adults moving to other regions, having interspecific competition acting on the death rates instead. Given that $R_0$ is as sensitive to the factors $d_{ii}$ and $d_{ij}$ as to the competition coefficients $c_{ii}$ and $c_{ij}$ (see Supporting Information), a more detailed view at the life stage at which competition is felt more strongly would contribute to a better understanding of the invasion risks in complex and interacting populations.

Looking at invasion risk with a focus on competition also allowed us to study a possible relationship between the competition in vectors and their potential feeding preferences. When a vector is under stronger competitive pressure, it is beneficial, from the point-of-view of the virus, that the vector more frequently bites a host that has a higher capability to transmit the virus. We saw how even very small host differences in disease susceptibility are sufficient to observe differences in $R_0$. One possible limitation of how we modeled this effect is that the parameters describing the preferences were limited to the interval [0, 2]. In reality, feeding preferences may be even stronger [39] or dependent on host availability (and thus affected by competition), so an approach that incorporates this effect in terms of rescaling the vector biting rates (as in for example [28]) could be considered in future studies.

The possible dilution effect of vector-borne diseases by non-competent species has been widely discussed [10, 40] and modeled [11, 41]. In tick-borne diseases, such as Lyme disease, non-competent hosts have been incorporated into models [42, 43]. In mosquito-borne diseases, such as WNV, similar efforts have been made [12, 38]. In any case, these non-competent hosts functioned simply as species that did not further transmit the pathogen, without consideration for how their ecological interactions with competent hosts may alter the transmission landscape. We included non-competent hosts, as well as non-competent vectors, that are able to compete with competent species. Non-competent vectors can have a diluting effect by competing with competent mosquitoes at the larval stage, reducing their abundances. Non-competent hosts can have a diluting effect by absorbing bites that could have happened instead on competent hosts, but their competition decreases the abundance of competent hosts, indirectly affecting $R_0$. In our simulations, the effect of dilution effect was greater than that of the non-competent host competition, resulting in lower values for $R_0$ compared with the system with competent hosts only. Under these conditions, the addition of extra species can have a beneficial effect by reducing the probability of an outbreak, even in cases where these species may decrease the number of competent hosts which normally could lead to a higher risk of invasion. The balance between the lack of competence, which has a diluting effect, and the competitive burden, which has an amplifying effect, is highly dependent on the epidemiological features of the system at study and more research is needed to address these questions. Further,

we chose to model non-competent hosts as dead-end hosts. If, besides not being infectious, they could not even ever become infected, one could alternatively regard species that are not susceptible to the pathogen but are being bitten by the vector. From the point of view of the pathogen, there is no difference between 'wasted' bites and 'completely wasted bites'. From the point of view of the vector, there is a difference if the dead-end hosts would have a high death rate, depending on vector preference and relative abundance of the various non-competent types. In our setup, these circumstances are assumed not to occur, and hence dead-end hosts are representative of non-competent hosts. Including species that do not attract or feed the vector, yet indirectly influence pathogen dynamics through their ecological interactions, would be an additional step. However, they are less likely to compete with the host species if they occupy different trophic levels and their effect would mostly be relevant if the vector feeding options were limited.

The $R_0$ values obtained in the simulations are dependent not only on the model and its assumptions, but also on the parameter values collected from the literature. If other ecological and epidemiological characteristics for the mosquito, competent and non-competent species, were used, the absolute estimates of $R_0$ will differ. For example, the carrying capacities selected (given by the intra-specific competition terms) have a direct effect not only on the species abundances, but also on the range of possible values for the competition coefficients, and can hence affect the range of $R_0$ estimated for the system. The value of our analysis is not in providing better absolute estimates of $R_0$ for West Nile Virus, but rather in showing patterns in how the value of $R_0$ for a typical mosquito-borne pathogen system can be affected by broader epidemiological and ecological interactions. It is also important to note the difficulty of comparing parameter estimates between different vector-borne disease systems. Even if they share the same vector species, data are collected from very different locations, in different seasons, or from different sets of host species, causing broad ranges for some of the parameters that need to be estimated.

We have made several simplifying assumptions in light of our focus on the invasion phase. Our assumption about the absence of host density dependence in the vector growth rate will also influence this because of feeding limitations at low host abundances. Our model does not take into consideration parameters that depend on the time of the year or environmental and spatially heterogeneous factors such as temperature, humidity, and availability of water. Vectors are sensitive to seasonal and other environmental changes, influencing their abundance locally and on large spatial scales. One could expect vectors to compete even more during times of the year, such as dryer periods, when suitable locations used for egg-laying are limited. Also host abundance is typically seasonal because of seasonal reproduction and seasonal presence, such as the case of migratory birds. By ignoring seasonality, we basically assume that the patterns of change in $R_0$ that result from ecological interaction will be affected less by seasonal variation than the range of values that $R_0$ takes. $R_0$ is not defined here with time-varying ingredients. When variation is periodic, a mathematical definition has been described (see [44] for details). Alternatively, one could calculate $R_0$ for each separate month of the year, fixing ingredients within each month. It is unclear what would express invasion potential best because one should then also take into account variation in introduction of the pathogen. Another choice we made was ignoring a latency period before the individuals become infectious. Our reasoning for this choice was that, intuitively, a latency period would lead to overall lower values of $R_0$ but not likely to a qualitatively different picture of how $R_0$ depends on competition. To test this, we expanded the model to explore latency periods in vectors for our results in Fig 3. Including latency periods does not change the observed patterns of $R_0$ in a qualitative sense (see Supporting Information), although quantitatively, latency periods lead to lower values of $R_0$ overall. Finally, choosing how one models the transmission rates regarding host availability

can impact the results. If we are studying habitats with low density of hosts, density rather than frequency dependent transmission terms would be more appropriate. The limitations and consequences of such choices for invasion risk prediction have been discussed at length in [14]. In such settings, additional assumptions we made will need to be relaxed, notably the density dependence in vector life history.

West Nile Virus was chosen as a motivation for our simulation studies. However, this model can easily be adapted to other vector-borne diseases. Particularly, those that circulate in the same vector and host populations, such as the Usutu virus or the avian malaria parasite. More generally speaking, our understanding of invasion risks of these and other vector-borne diseases can be improved by considering ecological interactions. This work should be seen as a proof-of-concept on how we can extend a vector-borne disease model to account for more complex layers of species interactions and understanding their impacts on emergence of infections in ecosystems communities.

## Supporting information

**S1 File. Including latency periods in vector species.**
(PDF)

**S2 File. Relationship between competition, the overall vector-to-host ratio, and the basic reproduction number.**
(PDF)

**S3 File. Elasticity of the basic reproduction number to model parameters.**
(PDF)

## Author Contributions

**Conceptualization:** Afonso Dimas Martins, J. A. P. Heesterbeek.

**Formal analysis:** Afonso Dimas Martins, J. A. P. Heesterbeek.

**Investigation:** Afonso Dimas Martins, J. A. P. Heesterbeek.

**Methodology:** Afonso Dimas Martins, J. A. P. Heesterbeek.

**Project administration:** J. A. P. Heesterbeek.

**Resources:** Afonso Dimas Martins.

**Software:** Afonso Dimas Martins.

**Supervision:** Quirine ten Bosch, J. A. P. Heesterbeek.

**Validation:** Afonso Dimas Martins, J. A. P. Heesterbeek.

**Visualization:** Afonso Dimas Martins, Quirine ten Bosch, J. A. P. Heesterbeek.

**Writing – original draft:** Afonso Dimas Martins, J. A. P. Heesterbeek.

**Writing – review & editing:** Afonso Dimas Martins, Quirine ten Bosch, J. A. P. Heesterbeek.

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
