## [Decision Letter · Decision Letter 0]

6 Jun 2022

PONE-D-22-09208Exploring the influence of competition on arbovirus invasion risk in ecosystemsPLOS ONE

Dear Dr. Dimas Martins,

Thank you for submitting your manuscript to PLOS ONE. After careful consideration, we feel that it has merit but does not fully meet PLOS ONE’s publication criteria as it currently stands. Therefore, we invite you to submit a revised version of the manuscript that addresses the points raised during the review process.

Both reviewers and myself commend the authors for an interesting and well-articulated modeling study. Both reviewers have important suggestions for better justifying certain model assumptions, such as frequency-dependent transmission, density-independent birth of vectors, the lack of a latency/incubation period for vectors, and how host competence is modeled. Some of these assumptions may simply require more justification in the methods and articulation of alternatives in the discussion, whereas others may warrant additional sensitivity analyses or revised model structure. The reviewers also have helpful suggestions to provide more context on the host-pathogen system used to ground the model and for how to improve figure readability and interpretation. 

We look forward to receiving your revised manuscript.

Kind regards,

Daniel Becker

Academic Editor

PLOS ONE

Journal Requirements:

Reviewers' comments:

Reviewer's Responses to Questions

**Comments to the Author**

1. Is the manuscript technically sound, and do the data support the conclusions?

Reviewer #1: Yes

Reviewer #2: Yes

2. Has the statistical analysis been performed appropriately and rigorously? 

Reviewer #1: Yes

Reviewer #2: Yes

3. Have the authors made all data underlying the findings in their manuscript fully available?

Reviewer #1: No

Reviewer #2: Yes

4. Is the manuscript presented in an intelligible fashion and written in standard English?

Reviewer #1: Yes

Reviewer #2: Yes

5. Review Comments to the Author

Reviewer #1: Overall, this manuscript tackles an interesting problem and attempts to incorporate biologically important complexity into a disease model for WNV. I appreciate that in this work the authors chose to include important complexity into their model (multi-host/multi-vector species with intra- and interspecific competition and vector feeding preferences) to better understand WNV dynamics, and that at times other assumptions and simplifications needed to be made. However, my main suggestion is that these assumptions be better justified and their implications discussed. Along these lines, I also would have appreciated more information/clarification on WNV-specific biology which would add value to this manuscript for a wider readership.

Major comments:

1. Lines 18-28: Well-written justification for the study.

2. Lines 75-77: Assumption that birth of vectors is independent of host species abundance. This is a reasonable assumption when looking at short time scales of transmission, which could be applicable in this system under certain circumstances. However, because this assumption could heavily influence the results, I would like more discussion of why it is a valid choice and some addition to the discussion section of other potential outcomes if this assumption were not made.

3. Lines 86-88: Again, these could be reasonable assumptions given the questions you are asking, but some justification of these choices (even with just citations) would be helpful

4. Lines 89-95: Choosing host-frequency dependent transmission for the model is a crucial assumption that greatly impacts the results regarding the influence of host competition on R0. Frequency-dependence is a valid choice for this system, but choosing differently could have resulted in very different final results. I suggest adding a citation on these lines to support frequency dependence, and then elaborating in the discussion section the strengths and limitations of this approach compared to others (see Wonham et al Ecology Letters 2006 for in depth examination).

5. Table 1: This is the only place you mention the actual vector/host species you are using for your model (C. pipiens/molestus and the American crow/House finch). Given the complexities of the WNV system, I’d suggest a brief discussion of the biology of WNV in these chosen species somewhere in the methods and why these were chosen in particular.

6. Lines 170-171: Here the authors make the assumption that mosquitos do not bite other species when investigating the importance of vector feeding preference. However, this negates the base model assumption made in lines 75-77 and discussed above. Given that the influence of host competition forms the crux of the interesting results from this model extension, it seems inappropriate to layer these opposing assumptions on top of one another. I would suggest either very clear justification from the authors on why these apparently diametric choices are acceptable, or re-considering the base assumption that host abundance will not impact the vector population.

7. Figure 5: In general, I found this figure confusing and do not feel it added much value. Maybe it would be more useful as a supplementary figure for some readers. If the authors decide to keep it, please make the y axis for b and d begin at zero and add labeling for hosts vs vectors and intra vs interspecific competition like the labels in Figure 3.

Minor comments:

1. Line 3, edit to qualify the statement since isn’t true of all vectors, depending on how you define a vector: “Vectors such as mosquitos usually feed on animals as they often require a blood meal for completing their lifestyle”.

2. Figure 3: The authors stated that they had to choose different color scales for each panel due to the differences in R0, but in looking at the range of results it seems R0 ranges from 0.6-4 across all panels. I think a color scale could be chosen such that each panel was on the same scale and therefore the results were easier to understand within and between each panel. However, if with some experimentation the authors did not think this was possible then I would suggest at least using two distinct scales, one for between vectors and one for between hosts and making those two scales distinct (white to red or white to blue, for example) for visual clarity.

3. Lines 348-349: semantics, but a relationship is either linear or not; something being “more non-linear” is kind of meaningless as a standalone statement.

4. Line 359: I think this is a typo and should read “In the bottom right both prefer N4”.

Reviewer #2: The authors present a computational exploration of the effects that intra- and interspecific competition between host and vector species have on parasite invasion into an ecosystem (community) (R0). They developed a mathematical model that includes up to two hosts and two vectors (three if you count the scenario with a dead-ends host and vector), and were able to explore how varying various ecological parameters (e.g. competition coefficients and vector to host ratios) influenced R0. I found their presentation of the mathematics required for this study are very clear, understandable, and very well explained. I appreciate the generality of their approach but also their desire to apply their model to a real system (West Nile Virus). However, I have some comments on some aspects of the model, study, and manuscript in general:

- In their model, the authors assume no latency for host or vector species. While I can understand this for the host species (as relative to their lifespan their incubation period may be rather short), I believe the model should include a latent period for the vectors. As the latent period for vectors **could** represent a long proportion of the adult lifespan of the vector, all individuals that become infected may not reach the time of infectiousness (i.e. the time they are able to retransmit/pass on the parasite). However, for your model or explorations, there are multiple ways you could include this in your already developed model. For example, you could design make this binary process where either you assume the vector has a latent period or does not. At the very least this should be discussed in the discussion section as it is a common aspect of many vector-borne parasite models. Additionally, is there an extrinsic incubation period for the WNV system in hosts and/or vectors?

- The authors assume that births of vector species are independent of host species abundance (line 75-80) and vectors have other feeding opportunities to meet the requirements of their lifecycle. This is an understandable simplifying assumption. However, later in the manuscript (line 170-171) the authors state that this is a closed system and that the mosquito does not bite any other animal. This seems to be the opposite of the earlier assumption.

- I'm not sure why the authors assume that the "non-competent hosts" were dead-end hosts. I believe the authors had two options when modeling non-competent hosts either: 1) dead-end host approach, hosts are able to become infected but do not pass on the parasite or 2) completely wasted bites, hosts cannot become infected but are fed on by infected vectors essentially **completely** wasting that bite from the point of view of the parasite. Why was one chosen over the other? It seems that choosing the dead-end host approach would allow those infected individuals to be included in the expression (matrix) for R0. Perhaps, some discussion of this decision and how the other ("completely wasted bites" scenario) may play out would be useful.

- I found the discussion to be lacking in how these findings relate back to previous literature and our current understanding of how competition affects parasite transmission in vector-borne parasites. Reading through the discussion seemed to be largely a recap of the results with very little reference to the literature.

- One interesting aspect that was not discussed much in the discussion are differences in phenology between host and vector species. The current approach assumes that there are no differences in births (or deaths) through time. However, many host-vector systems have different degrees of phenological overlap between host species, vector species, and both. In particular, this could be important for the WNV system with migratory bird species and mosquito emergence. While I understand the focal topic of this paper is the effects of competition, a discussion of host and vector phenology should be included.

Minor comments

- I found it interesting that the authors refer to the hosts and vectors community (which is what their model represents) as an "ecosystem" rather than a "community".

- Some of the lines in the model diagram (Fig 1 and 2) are hard to see. In particular the grey, yellow and green lines.

- I appreciate the inclusion of horizontal transmission for the hosts (i.e. a direct transmission route). This increases the generality of the model that could easily be "turned off" by setting the beta parameter to 0. However, why was the transmission only considered to be intraspecific and not between different hosts (the authors briefly mention crows as an example)? Additionally, how common is this route in WNV transmission?

6. PLOS authors have the option to publish the peer review history of their article (what does this mean?). If published, this will include your full peer review and any attached files.

Reviewer #1: No

Reviewer #2: **Yes: **John Vinson

---

## [Author Response · Author response to Decision Letter 0]

12 Jul 2022

We have addressed every reviewer comments in the file Response to Reviewers. Thank you for your feedback.

---

## [Decision Letter · Decision Letter 1]

2 Aug 2022

PONE-D-22-09208R1Exploring the influence of competition on arbovirus invasion risk in ecosystemsPLOS ONE

Dear Dr. Dimas Martins,

Thank you for submitting your manuscript to PLOS ONE. After careful consideration, we feel that it has merit but does not fully meet PLOS ONE’s publication criteria as it currently stands. Therefore, we invite you to submit a revised version of the manuscript that addresses the points raised during the review process.

Both original reviewers feel the manuscript has been much improved and is close to ready for publication. Some additional suggestions are posed to better justify invasion risk and to better interpret results in the context of the broader literature in the discussion, alongside more minor suggestions.============================== Please submit your revised manuscript by Sep 16 2022 11:59PM. If you will need more time than this to complete your revisions, please reply to this message or contact the journal office at plosone@plos.org. Please include the following items when submitting your revised manuscript:A rebuttal letter that responds to each point raised by the academic editor and reviewer(s). You should upload this letter as a separate file labeled 'Response to Reviewers'.A marked-up copy of your manuscript that highlights changes made to the original version. You should upload this as a separate file labeled 'Revised Manuscript with Track Changes'.An unmarked version of your revised paper without tracked changes. You should upload this as a separate file labeled 'Manuscript'.If applicable, we recommend that you deposit your laboratory protocols in protocols.io to enhance the reproducibility of your results. Protocols.io assigns your protocol its own identifier (DOI) so that it can be cited independently in the future. For instructions see: https://journals.plos.org/plosone/s/submission-guidelines#loc-laboratory-protocols. Additionally, PLOS ONE offers an option for publishing peer-reviewed Lab Protocol articles, which describe protocols hosted on protocols.io. Read more information on sharing protocols at https://plos.org/protocols?utm_medium=editorial-email&utm_source=authorletters&utm_campaign=protocols.

We look forward to receiving your revised manuscript.

Kind regards,

Daniel Becker

Academic Editor

PLOS ONE

Journal Requirements:

Reviewers' comments:

Reviewer's Responses to Questions

**Comments to the Author**

1. If the authors have adequately addressed your comments raised in a previous round of review and you feel that this manuscript is now acceptable for publication, you may indicate that here to bypass the “Comments to the Author” section, enter your conflict of interest statement in the “Confidential to Editor” section, and submit your "Accept" recommendation.

Reviewer #1: (No Response)

Reviewer #2: All comments have been addressed

2. Is the manuscript technically sound, and do the data support the conclusions?

Reviewer #1: Yes

Reviewer #2: Yes

3. Has the statistical analysis been performed appropriately and rigorously? 

Reviewer #1: Yes

Reviewer #2: Yes

4. Have the authors made all data underlying the findings in their manuscript fully available?

Reviewer #1: Yes

Reviewer #2: Yes

5. Is the manuscript presented in an intelligible fashion and written in standard English?

Reviewer #1: Yes

Reviewer #2: Yes

6. Review Comments to the Author

Reviewer #1: The authors addressed most critiques of the reviewers thoroughly. Some additional comments below, but if the authors address these comments, at the discretion of the editor, I believe the manuscript would be ready for publication without another round of review.

Major comments:

1. On this reading, I realized that the authors focus on the idea of invasion risk, and thus make certain model assumptions around that, but never really discuss in the introduction why they focus on this point in an outbreak. Ideally the answer won’t be “because it is mathematically expedient”, but something tied to the interesting biology of the WNV system, or how they imagine this multi-host/multi-vector framework would be extended to other systems, etc. This doesn’t need to be a huge addition, just a sentence or two to guide the reader why this is an interesting focus when they first specify that they want to focus on invasion risk (around line 50).

2. In the first round of review the other reviewer made a comment that the discussion section was too focused on rehashing the findings of the manuscript and not connecting it to the broader literature or scientific implications. On this reading, I agree with that comment and I do not feel the authors have addressed it. While they have added some information and new references, it is all still in the context of their own findings. I would love to see some of the repetition from the results section cut down in the discussion section and instead more ties added to the broader field, implications of this study in the context of invasion risk in other systems, etc.

Minor comments:

1. Line 15: In response to a comment from the other reviewer, the authors felt the need to justify the use of the word “ecosystem” as opposed to “community”. At the end of the day, “ecosystem” is still not a very applicable term here because the authors are not considering abiotic contributors, and I don’t think this added line makes the justification well. The authors could just use “ecological interactions” when that is what they are trying to convey.

2. Line 19: “These competition forces” could be changed to “These competitive forces”

3. Line 22: “Birds compete the mostly” should likely be either “Birst compete mostly” or “Birds compete the most”

4. Line 51: Switches from present tense used to describe the manuscript in the previous two sentences to future tense “we will”. I personally prefer present tense here.

5. Line 65: “We regard” is a bit awkward; I would use the phrasing “We focus on” instead

6. Line 71-72: “A compartmental model…” passive voice. I would change to “we constructed”

7. Figure 1/2: The other reviewer commented on difficulty seeing the arrows in the figures. I still personally find the light grey hard to discern

8. Figure 6: Add labels on the figure denoting which panels are host vs. vector, similar to your other figures.

9. Lines 546-563: This was added to address my previous comment wanting more information on why these specific hosts and vectors were chosen. I like this explanation but I think it should be in the introduction or methods so that a reader is aware of it from the beginning.

Reviewer #2: I would like to thank the authors for their responses to each of the reviewer comments. They have sufficiently addressed each of my concerns.

7. PLOS authors have the option to publish the peer review history of their article (what does this mean?). If published, this will include your full peer review and any attached files.

Reviewer #1: No

Reviewer #2: **Yes: **John E. Vinson

---

## [Author Response · Author response to Decision Letter 1]

16 Sep 2022

We thank the reviewers for their critical reading of our manuscript and for their remaining comments. Our detailed responses can be found the in Response file.

---

## [Decision Letter · Decision Letter 2]

22 Sep 2022

Exploring the influence of competition on arbovirus invasion risk in communities

PONE-D-22-09208R2

Dear Dr. Dimas Martins,

We’re pleased to inform you that your manuscript has been judged scientifically suitable for publication and will be formally accepted for publication once it meets all outstanding technical requirements.

Kind regards,

Daniel Becker

Academic Editor

PLOS ONE

Additional Editor Comments (optional):

Reviewers' comments:

Reviewer's Responses to Questions

**Comments to the Author**

1. If the authors have adequately addressed your comments raised in a previous round of review and you feel that this manuscript is now acceptable for publication, you may indicate that here to bypass the “Comments to the Author” section, enter your conflict of interest statement in the “Confidential to Editor” section, and submit your "Accept" recommendation.

Reviewer #1: All comments have been addressed

2. Is the manuscript technically sound, and do the data support the conclusions?

Reviewer #1: Yes

3. Has the statistical analysis been performed appropriately and rigorously? 

Reviewer #1: Yes

4. Have the authors made all data underlying the findings in their manuscript fully available?

Reviewer #1: Yes

5. Is the manuscript presented in an intelligible fashion and written in standard English?

Reviewer #1: Yes

6. Review Comments to the Author

Reviewer #1: The authors have done a wonderful job creating a manuscript that will be of interest to a broad variety of readers. I have no further comments or revisions.

7. PLOS authors have the option to publish the peer review history of their article (what does this mean?). If published, this will include your full peer review and any attached files.

Reviewer #1: No

---

## [Editor Report · Acceptance letter]

28 Sep 2022

PONE-D-22-09208R2 

Exploring the influence of competition on arbovirus invasion risk in communities 

Dear Dr. Dimas Martins:

I'm pleased to inform you that your manuscript has been deemed suitable for publication in PLOS ONE. Congratulations! Your manuscript is now with our production department. 

Kind regards, 

on behalf of

Dr. Daniel Becker 

Academic Editor

PLOS ONE